# A Generalization of Submodular Cover via the Diminishing Return Property on the Integer Lattice

**Tasuku Soma**
The University of Tokyo
tasuku_soma@mist.i.u-tokyo.ac.jp

**Yuichi Yoshida**
National Institute of Informatics, and
Preferred Infrastructure, Inc.
yyoshida@nii.ac.jp

## Abstract

We consider a generalization of the submodular cover problem based on the concept of diminishing return property on the integer lattice. We are motivated by real scenarios in machine learning that cannot be captured by (traditional) submodular set functions. We show that the generalized submodular cover problem can be applied to various problems and devise a bicriteria approximation algorithm. Our algorithm is guaranteed to output a log-factor approximate solution that satisfies the constraints with the desired accuracy. The running time of our algorithm is roughly $O(n \log(nr) \log r)$, where $n$ is the size of the ground set and $r$ is the maximum value of a coordinate. The dependency on $r$ is exponentially better than the naive reduction algorithms. Several experiments on real and artificial datasets demonstrate that the solution quality of our algorithm is comparable to naive algorithms, while the running time is several orders of magnitude faster.

## 1 Introduction

A function $f : 2^S \to \mathbb{R}_+$ is called *submodular* if $f(X) + f(Y) \geq f(X \cup Y) + f(X \cap Y)$ for all $X, Y \subseteq S$, where $S$ is a finite ground set. An equivalent and more intuitive definition is by the *diminishing return property*: $f(X \cup \{s\}) - f(X) \geq f(Y \cup \{s\}) - f(Y)$ for all $X \subseteq Y$ and $s \in S \setminus Y$. In the last decade, the optimization of a submodular function has attracted particular interest in the machine learning community. One reason of this is that many real-world models naturally admit the diminishing return property. For example, document summarization [12, 13], influence maximization in viral marketing [7], and sensor placement [10] can be described with the concept of submodularity, and efficient algorithms have been devised by exploiting submodularity (for further details, refer to [8]).

A variety of proposed models in machine learning [4, 13, 18] boil down to the *submodular cover* problem [21]; for given monotone and nonnegative submodular functions $f, c : 2^S \to \mathbb{R}_+$, and $\alpha > 0$, we are to

$$\text{minimize } c(X) \qquad \text{subject to } f(X) \geq \alpha. \qquad (1)$$

Intuitively, $c(X)$ and $f(X)$ represent the cost and the quality of a solution, respectively. The objective of this problem is to find $X$ of minimum cost with the worst quality guarantee $\alpha$. Although this problem is NP-hard since it generalizes the set cover problem, a simple greedy algorithm achieves tight log-factor approximation and it practically performs very well.

The aforementioned submodular models are based on the submodularity of a set function, a function defined on $2^S$. However, we often encounter problems that cannot be captured by a set function. Let us give two examples:

**Sensor Placement:** Let us consider the following sensor placement scenario. Suppose that we have several types of sensors with various energy levels. We assume a simple trade-off between

information gain and cost. Sensors of a high energy level can collect a considerable amount of information, but we have to pay a high cost for placing them. Sensors of a low energy level can be placed at a low cost, but they can only gather limited information. In this scenario, we want to decide *which type* of sensor should be placed at each spot, rather than just deciding whether to place a sensor or not. Such a scenario is beyond the existing models based on submodular set functions.

**Optimal Budget Allocation:** A similar situation also arises in the optimal budget allocation problem [2]. In this problem, we want to allocate budget among ad sources so that (at least) a certain number of customers is influenced while minimizing the total budget. Again, we have to decide *how much* budget should be set aside for each ad source, and hence set functions cannot capture the problem.

We note that a function $f : 2^S \to \mathbb{R}_+$ can be seen as a function defined on a Boolean hypercube $\{0,1\}^S$. Then, the above real scenarios prompt us to generalize the submodularity and the diminishing return property to functions defined on the *integer lattice* $\mathbb{Z}_+^S$. The most natural generalization of the diminishing return property to a function $f : \mathbb{Z}_+^S \to \mathbb{R}_+$ is the following inequality:

$$f(\boldsymbol{x} + \chi_s) - f(\boldsymbol{x}) \geq f(\boldsymbol{y} + \chi_s) - f(\boldsymbol{y}) \tag{2}$$

for $\boldsymbol{x} \leq \boldsymbol{y}$ and $s \in S$, where $\chi_s$ is the $s$-th unit vector. If $f$ satisfies (2), then $f$ also satisfies the following *lattice submodular inequality*:

$$f(\boldsymbol{x}) + f(\boldsymbol{y}) \geq f(\boldsymbol{x} \vee \boldsymbol{y}) + f(\boldsymbol{x} \wedge \boldsymbol{y}) \tag{3}$$

for all $\boldsymbol{x}, \boldsymbol{y} \in \mathbb{Z}_+^S$, where $\vee$ and $\wedge$ are the coordinate-wise max and min operations, respectively. While the submodularity and the diminishing return property are equivalent for set functions, this is not the case for functions over the integer lattice; the diminishing return property (2) is stronger than the lattice submodular inequality (3). We say that $f$ is *lattice submodular* if $f$ satisfies (3), and if $f$ further satisfies (2) we say that $f$ is *diminishing return submodular* (*DR-submodular* for short). One might feel that the DR-submodularity (2) is too restrictive. However, considering the fact that the diminishing return is more crucial in applications, we may regard the DR-submodularity (2) as the most natural generalization of the submodularity, at least for applications mentioned so far [17, 6]. For example, under a natural condition, the objective function in the optimal budget allocation satisfies (2) [17]. The DR-submodularity was also considered in the context of submodular welfare [6].

In this paper, we consider the following generalization of the submodular cover problem for set functions: Given a monotone DR-submodular function $f : \mathbb{Z}_+^S \to \mathbb{R}_+$, a subadditive function $c : \mathbb{Z}_+^S \to \mathbb{R}_+$, $\alpha > 0$, and $r \in \mathbb{Z}_+$, we are to

$$\text{minimize } c(\boldsymbol{x}) \qquad \text{subject to } f(\boldsymbol{x}) \geq \alpha, \quad \boldsymbol{0} \leq \boldsymbol{x} \leq r\boldsymbol{1}, \tag{4}$$

where we say that $c$ is *subadditive* if $c(\boldsymbol{x} + \boldsymbol{y}) \leq c(\boldsymbol{x}) + c(\boldsymbol{y})$ for all $\boldsymbol{x}, \boldsymbol{y} \in \mathbb{Z}_+^S$. We call problem (4) the *DR-submodular cover problem*. This problem encompasses problems that boil down to the submodular cover problem for set functions and their generalizations to the integer lattice. Furthermore, the cost function $c$ is generalized to a subadditive function. In particular, we note that two examples given above can be rephrased using this problem (see Section 4 for details).

If $c$ is also monotone DR-submodular, one can reduce the problem (4) to the set version (1) (for technical details, see Section 3.1). The problem of this naive reduction is that it only yields a *pseudo-polynomial time algorithm*; the running time depends on $r$ rather than $\log r$. Since $r$ can be huge in many practical settings (e.g., the maximum energy level of a sensor), even linear dependence on $r$ could make an algorithm impractical. Furthermore, for a general subadditive function $c$, this naive reduction does not work.

## 1.1 Our Contribution

For the problem (4), we devise a *bicriteria* approximation algorithm based on the decreasing threshold technique of [3]. More precisely, our algorithm takes the additional parameters $0 < \epsilon, \delta < 1$. The output $\boldsymbol{x} \in \mathbb{Z}_+^S$ of our algorithm is guaranteed to satisfy that $c(\boldsymbol{x})$ is at most $(1 + 3\epsilon)\rho \left(1 + \log \frac{d}{\beta}\right)$ times the optimum and $f(\boldsymbol{x}) \geq (1 - \delta)\alpha$, where $\rho$ is the curvature of $c$ (see Section 3 for the definition), $d = \max_s f(\chi_s)$ is the maximum value of $f$ over all standard unit vectors, and $\beta$ is the minimum value of the positive increments of $f$ in the feasible region.

**Running Time (dependency on $r$):** An important feature of our algorithm is that the running time depends on the bit length of $r$ only *polynomially* whereas the naive reduction algorithms depend on it *exponentially* as mentioned above. More precisely, the running time of our algorithm is $O(\frac{n}{\epsilon} \log \frac{nrc_{\max}}{\delta c_{\min}} \log r)$, which is polynomial in the input size, whereas the naive algorithm is only psuedo-polynomial time algorithm. In fact, our experiments using real and synthetic datasets show that our algorithm is considerably faster than naive algorithms. Furthermore, in terms of the objective value (that is, the cost of the output), our algorithm also exhibits comparable performance.

**Approximation Guarantee:** Our approximation guarantee on the cost is almost tight. Note that the DR submodular cover problem (4) includes the set cover problem, in which we are given a collection of sets, and we want to find a minimum number of sets that covers all the elements. In our context, $S$ corresponds to the collection of sets, the cost $c$ is the number of chosen sets, and $f$ is the number of covered elements. It is known that we cannot obtain an $o(\log m)$-approximation unless P $\neq$ NP, where $m$ is the number of elements [16]. However, since for the set cover problem we have $\rho = 1$, $d = O(m)$, and $\beta = 1$, our approximation guarantee is $O(\log m)$.

## 1.2   Related Work

Our result can be compared with several results in the literature for the submodular cover problem for set functions. It is shown by Wolsey [21] that if $c(X) = |X|$, a simple greedy algorithm yields $(1 + \log \frac{d}{\beta})$-approximation, which coincides with our approximation ratio except for the $(1 + 3\epsilon)$ factor. Note that $\rho = 1$ when $c(X) = |X|$, or more generally, when $c$ is modular. Recently, Wan et al. [20] discussed a slightly different setting, in which $c$ is also submodular and both $f$ and $c$ are integer valued. They proved that the greedy algorithm achieves $\rho H(d)$-approximation, where $H(d) = 1 + 1/2 + \cdots + 1/d$ is the $d$-th harmonic number. Again, their ratio asymptotically coincides with our approximation ratio (Note that $\beta \geq 1$ when $f$ is integer valued).

Another common submodular-based model in machine learning is in the form of the *submodular maximization problem*: Given a monotone submodular set function $f : \{0,1\}^S \to \mathbb{R}_+$ and a feasible set $P \subseteq [0,1]^S$ (e.g., a matroid polytope or a knapsack polytope), we want to maximize $f(\boldsymbol{x})$ subject to $\boldsymbol{x} \in P \cap \{0,1\}^S$. Such models can be widely found in various tasks as already described. We note that the submodular cover problem and the submodular maximization problem are somewhat dual to each other. Indeed, Iyer and Bilmes [5] showed that a bicriteria algorithm of one of these problems yields a bicriteria algorithm for the other. Being parallel to our setting, generalizing the submodular maximization problem to the integer lattice $\mathbb{Z}_+^S$ is a natural question. In this direction, Soma et al. [17] considered the maximization of lattice submodular functions (not necessarily being DR-submodular) and devised a constant-factor approximation pseudo-polynomial time algorithm. We note that our result is not implied by [17] via the duality of [5]. In fact, such reduction only yields a pseudo-polynomial time algorithm.

## 1.3   Organization of This Paper

The rest of this paper is organized as follows: Section 2 sets the mathematical basics of submodular functions over the integer lattice. Section 3 describes our algorithm and the statement of our main theorem. In Section 4, we show various experimental results using real and artificial datasets. Section 5 sketches the proof of the main theorem. Finally, we conclude the paper in Section 6.

## 2   Preliminaries

Let $S$ be a finite set. For each $s \in S$, we denote the $s$-th unit vector by $\chi_s$; that is, $\chi_s(t) = 1$ if $t = s$, otherwise $\chi_s(t) = 0$. A function $f : \mathbb{Z}^S \to \mathbb{R}$ is said to be *lattice submodular* if $f(\boldsymbol{x}) + f(\boldsymbol{y}) \geq f(\boldsymbol{x} \vee \boldsymbol{y}) + f(\boldsymbol{x} \wedge \boldsymbol{y})$ for all $\boldsymbol{x}, \boldsymbol{y} \in \mathbb{Z}^S$. A function $f$ is *monotone* if $f(\boldsymbol{x}) \geq f(\boldsymbol{y})$ for all $\boldsymbol{x}, \boldsymbol{y} \in \mathbb{Z}^S$ with $\boldsymbol{x} \geq \boldsymbol{y}$. For $\boldsymbol{x}, \boldsymbol{y} \in \mathbb{Z}^S$ and a function $f : \mathbb{Z}^S \to \mathbb{R}$, we denote $f(\boldsymbol{y} \mid \boldsymbol{x}) := f(\boldsymbol{y} + \boldsymbol{x}) - f(\boldsymbol{x})$. A function $f$ is *diminishing return submodular (or DR-submodular)* if $f(\boldsymbol{x} + \chi_s) - f(\boldsymbol{x}) \geq f(\boldsymbol{y} + \chi_s) - f(\boldsymbol{y})$ for each $\boldsymbol{x} \leq \boldsymbol{y} \in \mathbb{Z}^S$ and $s \in S$. For a DR-submodular function $f$, one can immediately check that $f(k\chi_s \mid \boldsymbol{x}) \geq f(k\chi_s \mid \boldsymbol{y})$ for arbitrary $\boldsymbol{x} \leq \boldsymbol{y}$, $s \in S$, and $k \in \mathbb{Z}_+$. A function $f$ is *subadditive* if $f(\boldsymbol{x} + \boldsymbol{y}) \leq f(\boldsymbol{x}) + f(\boldsymbol{y})$ for $\boldsymbol{x}, \boldsymbol{y} \in \mathbb{Z}^S$. For each $\boldsymbol{x} \in \mathbb{Z}_+^S$, we define $\{\boldsymbol{x}\}$ to be the multiset in which each $s \in S$ is contained $\boldsymbol{x}(s)$ times.

In [17], a lattice submodular function $f : \mathbb{Z}^S \to \mathbb{R}$ is said to have the diminishing return property if $f$ is *coordinate-wise concave*: $f(\boldsymbol{x} + 2\chi_s) - f(\boldsymbol{x} + \chi_s) \leq f(\boldsymbol{x} + \chi_s) - f(\boldsymbol{x})$ for each $\boldsymbol{x} \in \mathbb{Z}^S$ and $s \in S$. We note that our definition is consistent with [17]. Formally, we have the following lemma, whose proof can be found in Appendix.

**Lemma 2.1.** *A function $f : \mathbb{Z}^S \to \mathbb{R}$ is DR-submodular if and only if $f$ is lattice submodular and coordinate-wise concave.*

The following is fundamental for a monotone DR-submodular function. A proof is placed in Appendix due to the limitation of space.

**Lemma 2.2.** *For a monotone DR-submodular function $f$, $f(\boldsymbol{x}) - f(\boldsymbol{y}) \leq \sum_{s \in \{\boldsymbol{x}\}} f(\chi_s \mid \boldsymbol{y})$ for arbitrary $\boldsymbol{x}, \boldsymbol{y} \in \mathbb{Z}^S$.*

# 3 Algorithm for the DR-submodular Cover

Recall the DR-submodular cover problem (4). Let $f : \mathbb{Z}_+^S \to \mathbb{R}_+$ be a monotone DR-submodular function and let $c : \mathbb{Z}_+^S \to \mathbb{R}_+$ be a subadditive cost function. The objective is to minimize $c(\boldsymbol{x})$ subject to $f(\boldsymbol{x}) \geq \alpha$ and $\boldsymbol{0} \leq \boldsymbol{x} \leq r\boldsymbol{1}$, where $\alpha > 0$ and $r \in \mathbb{Z}_+$ are the given constants. Without loss of generality, we can assume that $\max\{f(\boldsymbol{x}) : \boldsymbol{0} \leq \boldsymbol{x} \leq r\boldsymbol{1}\} = \alpha$ (otherwise, we can consider $\widehat{f}(\boldsymbol{x}) := \min\{f(\boldsymbol{x}), \alpha\}$ instead of $f$). Furthermore, we can assume $c(\boldsymbol{x}) > 0$ for any $\boldsymbol{x} \in \mathbb{Z}_+^S$.

A pseudocode description of our algorithm is presented in Algorithm 1. The algorithm can be viewed as a modified version of the greedy algorithm and works as follows: We start with the initial solution $\boldsymbol{x} = \boldsymbol{0}$ and increase each coordinate of $\boldsymbol{x}$ gradually. To determine the amount of increments, the algorithm maintains a threshold $\theta$ that is initialized to be sufficiently large enough. For each $s \in S$, the algorithm finds the largest integer step size $0 < k \leq r - \boldsymbol{x}(s)$ such that the marginal cost-gain ratio $\frac{f(k\chi_s \mid \boldsymbol{x})}{kc(\chi_s)}$ is above the threshold $\theta$. If such $k$ exists, the algorithm updates $\boldsymbol{x}$ to $\boldsymbol{x} + k\chi_s$. After repeating this for each $s \in S$, the algorithm decreases the threshold $\theta$ by a factor of $(1 - \epsilon)$. If $\boldsymbol{x}$ becomes feasible, the algorithm returns the current $\boldsymbol{x}$. Even if $\boldsymbol{x}$ does not become feasible, the final $\boldsymbol{x}$ satisfies $f(\boldsymbol{x}) \geq (1 - \delta)\alpha$ if we iterate until $\theta$ gets sufficiently small.

---

**Algorithm 1** Decreasing Threshold for the DR-Submodular Cover Problem

**Input:** $f : \mathbb{Z}_+^S \to \mathbb{R}_+$, $c : \mathbb{Z}_+^S \to \mathbb{R}_+$, $r \in \mathbb{N}$, $\alpha > 0$, $\epsilon > 0$, $\delta > 0$.
**Output:** $\boldsymbol{0} \leq \boldsymbol{x} \leq r\boldsymbol{1}$ such that $f(\boldsymbol{x}) \geq \alpha$.
1: $\boldsymbol{x} \leftarrow \boldsymbol{0}$, $d \leftarrow \max\limits_{s \in S} f(\chi_s)$, $c_{\min} \leftarrow \min\limits_{s \in S} c(\chi_s)$, $c_{\max} \leftarrow \max\limits_{s \in S} c(\chi_s)$
2: **for** $(\theta = \frac{d}{c_{\min}}; \theta \geq \frac{\delta}{nc_{\max}r}d; \theta \leftarrow \theta(1 - \epsilon))$ **do**
3:     **for all** $s \in S$ **do**
4:         Find maximum integer $0 < k \leq r - \boldsymbol{x}(s)$ such that $\frac{f(k\chi_s \mid \boldsymbol{x})}{kc(\chi_s)} \geq \theta$ with binary search.
5:         **If** such $k$ exists **then** $\boldsymbol{x} \leftarrow \boldsymbol{x} + k\chi_s$.
6:         **If** $f(\boldsymbol{x}) \geq \alpha$ **then** break the outer **for** loop.
7: **return** $\boldsymbol{x}$

---

Before we claim the theorem, we need to define several parameters on $f$ and $c$. Let $\beta := \min\{f(\chi_s \mid \boldsymbol{x}) : s \in S, \boldsymbol{x} \in \mathbb{Z}_+^S, f(\chi_s \mid \boldsymbol{x}) > 0\}$ and $d := \max_s f(\chi_s)$. Let $c_{\max} := \max_s c(\chi_s)$ and $c_{\min} := \min_s c(\chi_s)$. Define the *curvature* of $c$ to be

$$\rho := \min_{\boldsymbol{x}^* : \text{optimal solution}} \frac{\sum_{s \in \{\boldsymbol{x}^*\}} c(\chi_s)}{c(\boldsymbol{x}^*)}. \tag{5}$$

**Definition 3.1.** *For $\gamma \geq 1$ and $0 < \delta < 1$, a vector $\boldsymbol{x} \in \mathbb{Z}_+^S$ is a $(\gamma, \delta)$-bicriteria approximate solution if $c(\boldsymbol{x}) \leq \gamma \cdot c(\boldsymbol{x}^*)$, $f(\boldsymbol{x}) \geq (1 - \delta)\alpha$, and $\boldsymbol{0} \leq \boldsymbol{x} \leq r\boldsymbol{1}$.*

Our main theorem is described below. We sketch the proof in Section 5.

**Theorem 3.2.** *Algorithm 1 outputs a $\left((1 + 3\epsilon)\rho\left(1 + \log\frac{d}{\beta}\right), \delta\right)$-bicriteria approximate solution in $O\left(\frac{n}{\epsilon} \log \frac{nrc_{\max}}{\delta c_{\min}} \log r\right)$ time.*

### 3.1 Discussion

**Integer-valued Case.** Let us make a simple remark on the case that $f$ is integer valued. Without loss of generality, we can assume $\alpha \in \mathbb{Z}_+$. Then, Algorithm 1 always returns a feasible solution for any $0 < \delta < 1/\alpha$. Therefore, our algorithm can be easily modified to an approximation algorithm if $f$ is integer valued.

**Definition of Curvature.** Several authors [5, 19] use a different notion of curvature called the *total curvature*, whose natural extension for a function over the integer lattice is as follows: The total curvature $\kappa$ of $c : \mathbb{Z}_+^S \to \mathbb{R}_+$ is defined as $\kappa := 1 - \min_{s \in S} \frac{c(\chi_s | r\mathbf{1} - \chi_s)}{c(\chi_s)}$. Note that $\kappa = 0$ if $c$ is modular, while $\rho = 1$ if $c$ is modular. For example, Iyer and Bilmes [5] devised a bicriteria approximation algorithm whose approximation guarantee is roughly $O((1 - \kappa)^{-1} \log \frac{\beta}{d})$.

Let us investigate the relation between $\rho$ and $\kappa$ for DR-submodular functions. One can show that $1 - \kappa \le \rho \le (1 - \kappa)^{-1}$ (see Lemma E.1 in Appendix), which means that our bound in terms of $\rho$ is tighter than one in terms of $(1 - \kappa)^{-1}$.

**Comparison to Naive Reduction Algorithm.** If $c$ is also a monotone DR-submodular function, one can reduce (4) to the set version (1) as follows. For each $s \in S$, create $r$ copies of $s$ and let $\tilde{S}$ be the set of these copies. For $\tilde{X} \subseteq \tilde{S}$, define $\boldsymbol{x}_{\tilde{X}} \in \mathbb{Z}_+^S$ be the integral vector such that $\boldsymbol{x}_{\tilde{X}}(s)$ is the number of copies of $s$ contained in $\tilde{X}$. Then, $\tilde{f}(\tilde{X}) := f(\boldsymbol{x}_{\tilde{X}})$ is submodular. Similarly, $\tilde{c}(\tilde{X}) := c(\boldsymbol{x}_{\tilde{X}})$ is also submodular if $c$ is a DR-submodular function. Therefore we may apply a standard greedy algorithm of [20, 21] to the reduced problem and this is exactly what Greedy does in our experiment (see Section 4). However, this straightforward reduction only yields a pseudo-polynomial time algorithm since $|\tilde{S}| = nr$; even if the original algorithm was linear, the resulting algorithm would require $O(nr)$ time. Indeed this difference is not negligible since $r$ can be quite large in practical applications, as illustrated by our experimental evaluation.

**Lazy Evaluation.** We finally note that we can combine the lazy evaluation technique [11, 14], which significantly reduces runtime in practice, with our algorithm. Specifically, we first push all the elements in $S$ to a max-based priority queue. Here, the key of an element $s \in S$ is $\frac{f(\chi_s)}{c(\chi_s)}$. Then the inner loop of Algorithm 1 is modified as follows: Instead of checking all the elements in $S$, we pop elements whose keys are at least $\theta$. For each popped element $s \in S$, we find $k$ such that $0 < k \le r - \boldsymbol{x}(s)$ with $\frac{f(k\chi_s | \boldsymbol{x})}{kc(\chi_s)} \ge \theta$ with binary search. If there is such $k$, we update $\boldsymbol{x}$ with $\boldsymbol{x} + k\chi_s$. Finally, we push $s$ again with the key $\frac{f(\chi_s | \boldsymbol{x})}{c(\chi_s)}$ if $\boldsymbol{x}(s) < r$.

The correctness of this technique is obvious because of the DR-submodularity of $f$. In particular, the key of each element $s \in S$ in the queue is always at least $\frac{f(\chi_s | \boldsymbol{x})}{c(\chi_s)}$, where $\boldsymbol{x}$ is the current vector. Hence, we never miss $s \in S$ with $\frac{f(k\chi_s | \boldsymbol{x})}{kc(\chi_s)} \ge \theta$.

## 4 Experiments

### 4.1 Experimental Setting

We conducted experiments on a Linux server with an Intel Xeon E5-2690 (2.90 GHz) processor and 256 GB of main memory. The experiments required, at most, 4 GB of memory. All the algorithms were implemented in C++ and compiled with g++ 4.6.3.

In our experiments, the cost function $c : \mathbb{Z}_+^S \to \mathbb{R}_+$ is always chosen as $c(\boldsymbol{x}) = \|\boldsymbol{x}\|_1 := \sum_{s \in S} \boldsymbol{x}(s)$. Let $f : \mathbb{Z}_+^S \to \mathbb{R}_+$ be a submodular function and $\alpha$ be the worst quality guarantee. We implemented the following four methods:

- Decreasing-threshold is our method with the lazy evaluation technique. We chose $\delta = 0.01$ as stated otherwise.

- Greedy is a method in which, starting from $\boldsymbol{x} = \mathbf{0}$, we iteratively increment $\boldsymbol{x}(s)$ for $s \in S$ that maximizes $f(\boldsymbol{x} + \chi_s) - f(\boldsymbol{x})$ until we get $f(\boldsymbol{x}) \ge \alpha$. We also implemented the lazy evaluation technique [11].

- **Degree** is a method in which we assign $\boldsymbol{x}(s)$ a value proportional to the marginal $f(\chi_s) - f(\boldsymbol{0})$, where $\|\boldsymbol{x}\|_1$ is determined by binary search so that $f(\boldsymbol{x}) \geq \alpha$. Precisely speaking, $\boldsymbol{x}(s)$ is approximately proportional to the marginal since $\boldsymbol{x}(s)$ must be an integer.
- **Uniform** is a method that returns $k\mathbf{1}$ for minimum $k \in \mathbb{Z}_+$ such that $f(k\mathbf{1}) \geq \alpha$.

We use the following real-world and synthetic datasets to confirm the accuracy and efficiency of our method against other methods. We set $r = 100,000$ for both problems.

**Sensor placement.** We used a dataset acquired by running simulations on a 129-vertex sensor network used in Battle of the Water Sensor Networks (BWSN) [15]. We used the "bwsn-utilities" [1] program to simulate 3000 random injection events to this network for a duration of 96 hours. Let $S$ and $E$ be the set of the 129 sensors in the network and the set of the 3000 events, respectively. For each sensor $s \in S$ and event $e \in E$, a value $z(s, e)$ is provided, which denotes the time, in minutes, the pollution has reached $s$ after the injection time.[1]

We define a function $f : \mathbb{Z}_+^S \to \mathbb{R}_+$ as follows: Let $\boldsymbol{x} \in \mathbb{Z}_+^S$ be a vector, where we regard $\boldsymbol{x}(s)$ as the energy level of the sensor $s$. Suppose that when the pollution reaches a sensor $s$, the probability that we can detect it is $1 - (1 - p)^{\boldsymbol{x}(s)}$, where $p = 0.0001$. In other words, by spending unit energy, we obtain an extra chance of detecting the pollution with probability $p$. For each event $e \in E$, let $s_e$ be the first sensor where the pollution is detected in that injection event. Note that $s_e$ is a random variable. Let $z_\infty = \max_{e \in E, s \in S} z(s, e)$. Then, we define $f$ as follows:

$$f(\boldsymbol{x}) = \mathop{\mathbf{E}}_{e \in E} \mathop{\mathbf{E}}_{s_e} [z_\infty - z(s_e, e)],$$

where $z(s_e, e)$ is defined as $z_\infty$ when there is no sensor that managed to detect the pollution. Intuitively speaking, $\mathop{\mathbf{E}}_{s_e}[z_\infty - z(s_e, e)]$ expresses how much time we managed to save in the event $e$ on average. Then, we take the average over all the events. A similar function was also used in [11] to measure the performance of a sensor allocation although they only considered the case $p = 1$. This corresponds to the case that by spending unit energy at a sensor $s$, we can always detect the pollution that has reached $s$. We note that $f(\boldsymbol{x})$ is DR-submodular (see Lemma F.1 for the proof).

**Budget allocation problem.** In order to observe the behavior of our algorithm for large-scale instances, we created a synthetic instance of the budget allocation problem [2, 17] as follows: The instance can be represented as a bipartite graph $(S, T; E)$, where $S$ is a set of 5,000 vertices and $T$ is a set of 50,000 vertices. We regard a vertex in $S$ as an ad source, and a vertex in $T$ as a person. Then, we fix the degrees of vertices in $S$ so that their distribution obeys the power law of $\gamma := 2.5$; that is, the fraction of ad sources with out-degree $d$ is proportional to $d^{-\gamma}$. For a vertex $s \in S$ of the supposed degree $d$, we choose $d$ vertices in $T$ uniformly at random and connect them to $s$ with edges. We define a function $f : \mathbb{Z}_+^S \to \mathbb{R}_+$ as

$$f(\boldsymbol{x}) = \sum_{t \in T} \Big( 1 - \prod_{s \in \Gamma(t)} (1 - p)^{\boldsymbol{x}(s)} \Big), \tag{6}$$

where $\Gamma(t)$ is the set of vertices connected to $t$ and $p = 0.0001$. Here, we suppose that, by investing a unit cost to an ad source $s \in S$, we have an extra chance of influencing a person $t \in T$ with $s \in \Gamma(t)$ with probability $p$. Then, $f(\boldsymbol{x})$ can be seen as the expected number of people influenced by ad sources. We note that $f$ is known to be a monotone DR-submodular function [17].

### 4.2 Experimental Results

Figure 1 illustrates the obtained objective value $\|\boldsymbol{x}\|_1$ for various choices of the worst quality guarantee $\alpha$ on each dataset. We chose $\epsilon = 0.01$ in Decreasing threshold. We can observe that Decreasing threshold attains almost the same objective value as Greedy, and it outperforms Degree and Uniform.

Figure 2 illustrates the runtime for various choices of the worst quality guarantee $\alpha$ on each dataset. We chose $\epsilon = 0.01$ in Decreasing threshold. We can observe that the runtime growth of Decreasing threshold is significantly slower than that of Greedy.

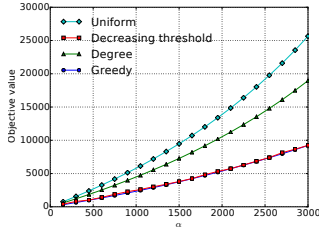

(a) Sensor placement (BWSN)

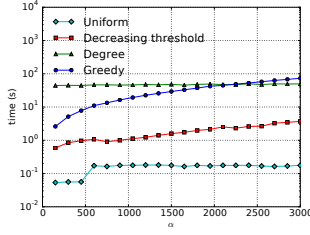

(a) Sensor placement (BWSN)

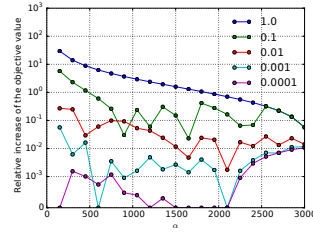

(a) Relative cost increase

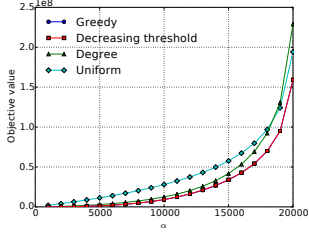

(b) Budget allocation (synthetic)

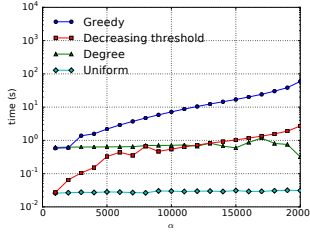

(b) Budget allocation (synthetic)

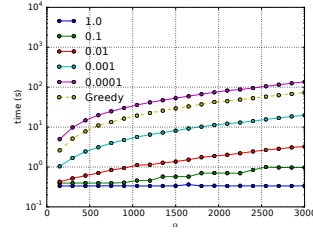

(b) Runtime

Figure 1: Objective values          Figure 2: Runtime          Figure 3: Effect of $\epsilon$

Figures 3(a) and 3(b) show the relative increase of the objective value and the runtime, respectively, of our method against Greedy on the BWSN dataset. We can observe that the relative increase of the objective value gets smaller as $\alpha$ increases. This phenomenon can be well explained by considering the extreme case that $\alpha = \max f(r\mathbf{1})$. In this case, we need to choose $\boldsymbol{x} = r\mathbf{1}$ anyway in order to achieve the worst quality guarantee, and the order of increasing coordinates of $\boldsymbol{x}$ does not matter. Also, we can see that the empirical runtime grows as a function of $\frac{1}{\epsilon}$, which matches our theoretical bound.

## 5 Proof of Theorem 3.2

In this section, we outline the proof of the main theorem. Proofs of some minor claims can be found in Appendix.

First, we introduce a notation. Let us assume that $\boldsymbol{x}$ is updated $L$ times in the algorithm. Let $\boldsymbol{x}_i$ be the variable $\boldsymbol{x}$ after the $i$-th update ($i = 0, \ldots, L$). Note that $\boldsymbol{x}_0 = \mathbf{0}$ and $\boldsymbol{x}_L$ is the final output of the algorithm. Let $s_i \in S$ and $k_i \in \mathbb{Z}_+$ be the pair used in the $i$-th update for $i = 1, \ldots, L$; that is, $\boldsymbol{x}_i = \boldsymbol{x}_{i-1} + k_i \chi_{s_i}$ for $i = 1, \ldots, L$. Let $\mu_0 := 0$ and $\mu_i := \frac{k_i c(\chi_{s_i})}{f(k_i \chi_{s_i} \mid \boldsymbol{x}_{i-1})}$ for $i = 1, \ldots, L$. Let $\hat{\mu}_0 := 0$ and $\hat{\mu}_i := \theta_i^{-1}$ for $i = 1, \ldots, L$, where $\theta_i$ is the threshold value on the $i$-th update. Note that $\hat{\mu}_{i-1} \le \hat{\mu}_i$ for $i = 1, \ldots, L$. Let $\boldsymbol{x}^*$ be an optimal solution such that $\rho \cdot c(\boldsymbol{x}^*) = \sum_{s \in \{\boldsymbol{x}^*\}} c(\chi_s)$.

We regard that in the $i$-th update, the elements of $\{\boldsymbol{x}^*\}$ are charged by the value of $\mu_i(f(\chi_s \mid \boldsymbol{x}_{i-1}) - f(\chi_s \mid \boldsymbol{x}_i))$. Then, the total charge on $\{\boldsymbol{x}^*\}$ is defined as

$$T(\boldsymbol{x}, f) := \sum_{s \in \{\boldsymbol{x}^*\}} \sum_{i=1}^{L} \mu_i(f(\chi_s \mid \boldsymbol{x}_{i-1}) - f(\chi_s \mid \boldsymbol{x}_i)).$$

**Claim 5.1.** *Let us fix $1 \le i \le L$ arbitrary and let $\theta$ be the threshold value on the $i$-th update. Then,*

$$\frac{f(k_i \chi_{s_i} \mid \boldsymbol{x}_{i-1})}{k_i c(\chi_{s_i})} \ge \theta \quad and \quad \frac{f(\chi_s \mid \boldsymbol{x}_{i-1})}{c(\chi_s)} \le \frac{\theta}{1 - \epsilon} \quad (s \in S).$$

Eliminating $\theta$ from the inequalities in Claim 5.1, we obtain

$$\frac{k_i c(\chi_{s_i})}{f(k_i \chi_{s_i} \mid \boldsymbol{x}_{i-1})} \le \frac{1}{1 - \epsilon} \frac{c(\chi_s)}{f(\chi_s \mid \boldsymbol{x}_{i-1})} \quad (i = 1, \ldots, L, \quad s \in S) \tag{7}$$

Furthermore, we have $\mu_i \le \hat{\mu}_i \le \frac{1}{1-\epsilon}\mu_i$ for $i = 1, \ldots, L$.

**Claim 5.2.** $c(\boldsymbol{x}) \le \frac{1}{1-\epsilon}T(\boldsymbol{x}, f)$.

**Claim 5.3.** *For each $s \in \{\boldsymbol{x}^*\}$, the total charge on $s$ is at most $\frac{1}{1-\epsilon}(1 + \log(d/\beta))c(\chi_s)$.*

*Proof.* Let us fix $s \in \{\boldsymbol{x}^*\}$ and let $l$ be the minimum $i$ such that $f(\chi_s \mid \boldsymbol{x}_i) = 0$. By (7), we have

$$\mu_i = \frac{k_i c(\chi_{s_i})}{f(k_i \chi_{s_i} \mid \boldsymbol{x}_{i-1})} \le \frac{1}{1-\epsilon} \cdot \frac{c(\chi_s)}{f(\chi_s \mid \boldsymbol{x}_{i-1})}. \qquad (i = 1, \ldots, l)$$

Then, we have

$$\sum_{i=1}^{L} \mu_i(f(\chi_s \mid \boldsymbol{x}_{i-1}) - f(\chi_s \mid \boldsymbol{x}_i)) = \sum_{i=1}^{l-1} \mu_i(f(\chi_s \mid \boldsymbol{x}_{i-1}) - f(\chi_s \mid \boldsymbol{x}_i)) + \mu_l f(\chi_s \mid \boldsymbol{x}_{l-1})$$

$$\le \frac{1}{1-\epsilon}c(\chi_s)\Big(\sum_{i=1}^{l-1} \frac{(f(\chi_s \mid \boldsymbol{x}_{i-1}) - f(\chi_s \mid \boldsymbol{x}_i))}{f(\chi_s \mid \boldsymbol{x}_{i-1})} + \frac{f(\chi_s \mid \boldsymbol{x}_{l-1})}{f(\chi_s \mid \boldsymbol{x}_{l-1})}\Big)$$

$$\le \frac{1}{1-\epsilon}c(\chi_s)\Big(1 + \sum_{i=1}^{l-1}\Big(1 - \frac{f(\chi_s \mid \boldsymbol{x}_i)}{f(\chi_s \mid \boldsymbol{x}_{i-1})}\Big)\Big)$$

$$\le \frac{1}{1-\epsilon}c(\chi_s)\Big(1 + \sum_{i=1}^{l-1}\log\frac{f(\chi_s \mid \boldsymbol{x}_{i-1})}{f(\chi_s \mid \boldsymbol{x}_i)}\Big) \qquad \text{(since } 1 - 1/x \le \log x \text{ for } x \ge 1\text{)}$$

$$= \frac{1}{1-\epsilon}c(\chi_s)\Big(1 + \log\frac{f(\chi_s \mid \boldsymbol{x}_0)}{f(\chi_s \mid \boldsymbol{x}_{l-1})}\Big) \le \frac{1}{1-\epsilon}\Big(1 + \log\frac{d}{\beta}\Big)c(\chi_s) \qquad \square$$

*Proof of Theorem 3.2.* Combining these claims, we have

$$c(\boldsymbol{x}) \le \frac{1}{1-\epsilon} \cdot T(\boldsymbol{x}, f) \le \frac{1}{(1-\epsilon)^2} \cdot \Big(1 + \log\frac{d}{\beta}\Big) \cdot \sum_{s \in \{\boldsymbol{x}^*\}} c(\chi_s) \le (1 + 3\epsilon) \cdot \Big(1 + \log\frac{d}{\beta}\Big) \cdot \rho c(\boldsymbol{x}^*).$$

Thus, $\boldsymbol{x}$ is an approximate solution with the desired ratio.

Let us see that $\boldsymbol{x}$ approximately satisfies the constraint; that is, $f(\boldsymbol{x}) \ge (1 - \delta)\alpha$. We will now consider a slightly modified version of the algorithm; in the modified algorithm, the threshold is updated until $f(\boldsymbol{x}) = \alpha$. Let $\boldsymbol{x}'$ be the output of the modified algorithm. Then, we have

$$f(\boldsymbol{x}') - f(\boldsymbol{x}) \le \sum_{s \in \{\boldsymbol{x}'\}} f(\chi_s \mid \boldsymbol{x}) \le \sum_{s \in \{\boldsymbol{x}'\}} \frac{\delta c(\chi_s)}{c_{\max}nr}d \le \delta d \le \delta\alpha$$

The third inequality holds since $c(\chi_s) \le c_{\max}$ and $|\{\boldsymbol{x}'\}| \le nr$. Thus $f(\boldsymbol{x}) \ge (1 - \delta)\alpha$. $\qquad \square$

## 6 Conclusions

In this paper, motivated by real scenarios in machine learning, we generalized the submodular cover problem via the diminishing return property over the integer lattice. We proposed a bicriteria approximation algorithm with the following properties: (i) The approximation ratio to the cost almost matches the one guaranteed by the greedy algorithm [21] and is almost tight in general. (ii) We can satisfy the worst solution quality with the desired accuracy. (iii) The running time of our algorithm is roughly $O(n \log n \log r)$. The dependency on $r$ is exponentially better than that of the greedy algorithm. We confirmed by experiment that compared with the greedy algorithm, the solution quality of our algorithm is almost the same and the runtime is several orders of magnitude faster.

#### Acknowledgments

The first author is supported by JSPS Grant-in-Aid for JSPS Fellows. The second author is supported by JSPS Grant-in-Aid for Young Scientists (B) (No. 26730009), MEXT Grant-in-Aid for Scientific Research on Innovative Areas (24106003), and JST, ERATO, Kawarabayashi Large Graph Project. The authors thank Satoru Iwata and Yuji Nakatsukasa for reading a draft of this paper.

## Footnotes

[1] Although three other values are provided, they showed similar empirical results and we omit them.

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
