[Supplementary Material]

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

. Let us take $\boldsymbol{x}$ and $\boldsymbol{y}$ with $\boldsymbol{x} \leq \boldsymbol{y}$, and fix $s \in S$. We can write $\boldsymbol{y} = \boldsymbol{x} + \sum_{t \in S^+} k_t \chi_t$, where $S^+ := \{t \in S : \boldsymbol{y}(t) > \boldsymbol{x}(t)\}$ and $k_t := \boldsymbol{y}(t) - \boldsymbol{x}(t)$ for each $t \in S^+$. If $s \notin S^+$, then

$$
\begin{aligned}
f(\boldsymbol{x} + \chi_s) &+ f(\boldsymbol{y}) \\
&\geq f((\boldsymbol{x} + \chi_s) \vee \boldsymbol{y}) + f((\boldsymbol{x} + \chi_s) \wedge \boldsymbol{y}) && \text{(by the lattice submodularity)} \\
&= f(\boldsymbol{y} + \chi_s) + f(\boldsymbol{x}), && \text{(since } \boldsymbol{x} \leq \boldsymbol{y} \text{ and } s \notin S^+)
\end{aligned}
$$

which implies $f(\chi_s \mid \boldsymbol{x}) \geq f(\chi_s \mid \boldsymbol{y})$. If $s \in S^+$, we have

$$
\begin{aligned}
f(\boldsymbol{x} + \chi_s) &- f(\boldsymbol{x}) \\
&\geq f(\boldsymbol{x} + k_s \chi_s + \chi_s) - f(\boldsymbol{x} + k_s \chi_s) && \text{(by the coordinate-wise concavity)} \\
&\geq f\left(\boldsymbol{x} + \sum_{t \in S^+} k_t \chi_t + \chi_s\right) - f\left(\boldsymbol{x} + \sum_{t \in S^+} k_t \chi_t\right) && \text{(by the lattice submodularity)} \\
&= f(\boldsymbol{y} + \chi_s) - f(\boldsymbol{y}). && \square
\end{aligned}
$$

(*Only if part*) Since the coordinate-wise concavity is immediate, we will show the lattice submodularity. Let us take $\boldsymbol{x}$ and $\boldsymbol{y}$ arbitrary, and let us write $\boldsymbol{y} = \boldsymbol{x} \wedge \boldsymbol{y} + \sum_{i=1}^l k_l \chi_{s_l}$, where $s_1, \ldots, s_l$ are all the elements $s$ of $S$ such that $\boldsymbol{y}(s) > \boldsymbol{x}(s)$ and $k_i := \boldsymbol{y}(s_i) - \boldsymbol{x}(s_i)$ for $i = 1, \ldots, l$. Then, we have $\boldsymbol{x} \vee \boldsymbol{y} = \boldsymbol{x} + \sum_{i=1}^l k_l \chi_{s_l}$. Using the DR-submodularity repeatedly, we obtain

$$
\begin{aligned}
f\left(\boldsymbol{x} + \sum_{i=1}^j k_i \chi_{s_i}\right) &- f\left(\boldsymbol{x} + \sum_{i=1}^{j-1} k_i \chi_{s_i}\right) \\
&\leq f\left(\boldsymbol{x} \wedge \boldsymbol{y} + \sum_{i=1}^j k_i \chi_{s_i}\right) - f\left(\boldsymbol{x} \wedge \boldsymbol{y} + \sum_{i=1}^{j-1} k_i \chi_{s_i}\right)
\end{aligned}
$$

for $j = 1, \ldots, l$. Summing up these inequalities yields $f(\boldsymbol{x} \vee \boldsymbol{y}) - f(\boldsymbol{x}) \leq f(\boldsymbol{y}) - f(\boldsymbol{x} \wedge \boldsymbol{y})$, which is the lattice submodular inequality.

# B  Proof of Lemma 2.2

*Proof.* Let us take $\boldsymbol{x}, \boldsymbol{y}$ arbitrary and write $\boldsymbol{x} \vee \boldsymbol{y} = \boldsymbol{y} + \sum_{i=1}^l k_i \chi_{s_i}$ as in the proof of Lemma 2.1. Then

$$
\begin{aligned}
f(\boldsymbol{x}) &- f(\boldsymbol{y}) \\
&\leq f(\boldsymbol{x} \vee \boldsymbol{y}) - f(\boldsymbol{y}) \\
&= \sum_{j=1}^l f\left(k_j \chi_{s_j} \,\middle|\, \boldsymbol{y} + \sum_{i=1}^{j-1} k_i \chi_{s_i})\right) \\
&\leq \sum_{j=1}^l f(k_j \chi_{s_j} \mid \boldsymbol{y}) && \text{(by Lemma 2.1)} \\
&\leq \sum_{j=1}^l k_j f(\chi_{s_j} \mid \boldsymbol{y}) && \text{(by the DR-submodularity)} \\
&\leq \sum_{s \in \{\boldsymbol{x}\}} f(\chi_s \mid \boldsymbol{y}) && \square
\end{aligned}
$$

# C  Proof of Claim 5.1

*Proof.* The first inequality is immediate from the fact that $(k_i, s_i)$ is chosen by the algorithm. For the second inequality, if $\theta = d/c_{\min}$ then it is trivial. Thus, we can assume $\theta < d/c_{\min}$; that is, there

were at least one threshold update. Suppose to the contrary that $\frac{f(\chi_s \mid \boldsymbol{x}_{i-1})}{c(\chi_s)} > \frac{\theta}{1-\epsilon}$ for some $s \in S$. Let $k$ be the chosen step size for the previous update on the $s$-th component and let $\boldsymbol{x}'$ be the variable $\boldsymbol{x}$ at that time. Then, $f((k+1)\chi_s \mid \boldsymbol{x}') \geq f(k\chi_s \mid \boldsymbol{x}') + f(\chi_s \mid \boldsymbol{x}) > \frac{\theta}{1-\epsilon}kc(\chi_s) + \frac{\theta}{1-\epsilon}c(\chi_s) = \frac{\theta}{1-\epsilon}(k+1)c(\chi_s)$, which contradicts the choice of $k$. $\qquad\square$

# D  Proof of Claim 5.2

*Proof.*

$$
\begin{aligned}
c(\boldsymbol{x}) &\leq \sum_{i=1}^{L} k_i c(\chi_{s_i}) && \text{(since } c \text{ is subadditive)} \\
&= \sum_{i=1}^{L} \mu_i f(k_i \chi_{s_i} \mid \boldsymbol{x}_{i-1}) = \sum_{i=1}^{L} \mu_i (f(\boldsymbol{x}_i) - f(\boldsymbol{x}_{i-1})) \\
&\leq \sum_{i=1}^{L} \hat{\mu}_i (f(\boldsymbol{x}_i) - f(\boldsymbol{x}_{i-1})) \\
&= \sum_{i=1}^{L} \hat{\mu}_i (f(\boldsymbol{x}^*) - f(\boldsymbol{x}_{i-1})) - \sum_{i=1}^{L} \hat{\mu}_i (f(\boldsymbol{x}^*) - f(\boldsymbol{x}_i)) \\
&= \sum_{i=1}^{L} \hat{\mu}_i (f(\boldsymbol{x}^*) - f(\boldsymbol{x}_{i-1})) - \sum_{i=1}^{L} \hat{\mu}_{i-1} (f(\boldsymbol{x}^*) - f(\boldsymbol{x}_{i-1})) \\
&&& \hspace{-6cm} \text{(since } \hat{\mu}_0 = 0 \text{ and } f(\boldsymbol{x}^*) = f(\boldsymbol{x}_L)) \\
&= \sum_{i=1}^{L} (\hat{\mu}_i - \hat{\mu}_{i-1})(f(\boldsymbol{x}^*) - f(\boldsymbol{x}_{i-1})) \leq \sum_{i=1}^{L} (\hat{\mu}_i - \hat{\mu}_{i-1}) \sum_{s \in \{\boldsymbol{x}^*\}} f(\chi_s \mid \boldsymbol{x}_{i-1}) \\
&= \sum_{i=1}^{L} (\hat{\mu}_i - \hat{\mu}_{i-1}) \sum_{s \in \{\boldsymbol{x}^*\}} f(\chi_s \mid \boldsymbol{x}_{i-1}) = \sum_{s \in \{\boldsymbol{x}^*\}} \sum_{i=1}^{L} (\hat{\mu}_i - \hat{\mu}_{i-1}) f(\chi_s \mid \boldsymbol{x}_{i-1}) \\
&= \sum_{s \in \{\boldsymbol{x}^*\}} \left( \sum_{i=1}^{L} \hat{\mu}_i f(\chi_s \mid \boldsymbol{x}_{i-1}) - \sum_{i=1}^{L} \hat{\mu}_{i-1} f(\chi_s \mid \boldsymbol{x}_{i-1}) \right) \\
&= \sum_{s \in \{\boldsymbol{x}^*\}} \sum_{i=1}^{L} \hat{\mu}_i \left( f(\chi_s \mid \boldsymbol{x}_{i-1}) - f(\chi_s \mid \boldsymbol{x}_i) \right) && \text{(since } \hat{\mu}_0 = 0 \text{ and } f(\boldsymbol{x}^*) = f(\boldsymbol{x}_L)) \\
&= (1-\epsilon)^{-1} T(\boldsymbol{x}, f) && \text{(since } \hat{\mu}_i \leq (1-\epsilon)^{-1} \mu_i)
\end{aligned}
$$

$\qquad\square$

# E  Relations between curvatures

**Lemma E.1.** *Let $c : \mathbb{Z}_+ \to \mathbb{R}_+$ be a DR-submodular function, and let $\rho$ and $\kappa$ be the curvature and the total curvature of $c$, respectively. Then, we have $1 - \kappa \leq \rho \leq (1-\kappa)^{-1}$.*

**Fact E.2.** *For $a_1, b_1, a_2, b_2, \ldots, a_n, b_n > 0$, $\frac{a_1+a_2+\cdots+a_n}{b_1+b_2+\cdots+b_n} \geq \min\left\{ \frac{a_1}{b_1}, \frac{a_2}{b_2}, \ldots, \frac{a_n}{b_n} \right\}$.*

*Proof.* Let $\boldsymbol{x}^*$ be the optimal solution of the submodular cover problem that attains the minimum of (5)

$$\rho = \frac{\sum_{s \in \{\boldsymbol{x}^*\}} c(\chi_s)}{c(\boldsymbol{x}^*)}$$

$$\geq \frac{\sum_{s \in \{\boldsymbol{x}^*\}} c(\chi_s \mid r\mathbf{1} - \chi_s)}{\sum_{s \in \{\boldsymbol{x}^*\}} c(\chi_s)} \quad \text{(since } c(\chi_s) \geq c(\chi_s \mid r\mathbf{1} - \chi_s) \text{ and } c(\boldsymbol{x}^*) \leq \sum_{s \in \{\boldsymbol{x}^*\}} c(\chi_s))$$

$$\geq \min_{s \in \{\boldsymbol{x}^*\}} \frac{c(\chi_s \mid r\mathbf{1} - \chi_s)}{c(\chi_s)} \quad \text{(by Fact E.2)}$$

$$\geq \min_{s \in S} \frac{c(\chi_s \mid r\mathbf{1} - \chi_s)}{c(\chi_s)} = 1 - \kappa.$$

On the other hand, let $\boldsymbol{x}^* = \chi_{s_1} + \cdots + \chi_{s_k}$, where $s_1, \ldots, s_k$ are the elements of the support of $\boldsymbol{x}^*$, $\boldsymbol{x}_i := \chi_{s_1} + \cdots + \chi_{s_i}$ $(i = 1, \ldots, k)$, and $\boldsymbol{x}_0 := \mathbf{0}$. Similar argument shows that

$$\frac{1}{\rho} = \frac{c(\boldsymbol{x}^*)}{\sum_{s \in \{\boldsymbol{x}^*\}} c(\chi_s)}$$

$$= \frac{\sum_{i=1}^{k} c(\chi_{s_i} \mid \boldsymbol{x}_{i-1})}{\sum_{s \in \{\boldsymbol{x}^*\}} c(\chi_s)}$$

$$\geq \frac{\sum_{i=1}^{k} c(\chi_{s_i} \mid r\mathbf{1} - \chi_{s_i})}{\sum_{s \in \{\boldsymbol{x}^*\}} c(\chi_s)} \quad \text{(since } \boldsymbol{x}_{i-1} \leq r\mathbf{1} - \chi_{s_i})$$

$$= \frac{\sum_{s \in \{\boldsymbol{x}^*\}} c(\chi_s \mid r\mathbf{1} - \chi_s)}{\sum_{s \in \{\boldsymbol{x}^*\}} c(\chi_s)} \geq 1 - \kappa.$$

Thus $\rho^{-1} \geq 1 - \kappa$, which is equivalent to $\rho \leq (1 - \kappa)^{-1}$. $\qquad\square$

# F  DR-submodularity of the sensor placement problem

We use notations used in the experiments on sensor placement of Section 4.

**Lemma F.1.** *The function $f : \mathbb{Z}_+^S \to \mathbb{R}_+$ is DR-submodular.*

*Proof.* Let us define $f_e : \mathbb{Z}_+^S \to \mathbb{R}_+$ as $f_e = \mathop{\mathbf{E}}_{s_e}[z_\infty - z(s_e, e)]$. It suffices to show that $f_e$ is DR-submodular since then $f = \mathop{\mathbf{E}}_{e \in E} f_e$ is also DR-submodular.

Let $\tilde{S}$ be the set obtained by copying each sensor $s \in S$ exactly $r$ times, and consider the function $\tilde{f}_e : 2^{\tilde{S}} \to \mathbb{R}_+$ given by the naive reduction (see Subsection 3.1). Then, we can regard $\tilde{f}(T)$ for $T \subseteq \tilde{S}$ as the expected cost when each sensor $s \in T$ tries to detect the pollution independently. Note that we have $f(\boldsymbol{x}) = \tilde{f}(T_{\boldsymbol{x}})$, where $T_{\boldsymbol{x}} \subseteq \tilde{S}$ contains $\boldsymbol{x}(s)$ copies of each sensor $s \in S$.

We note that $\tilde{f}$ satisfies the submodularity [9] (for set functions). Hence for $s \in S$ and $\boldsymbol{x} \leq \boldsymbol{y}$ with $\boldsymbol{y}(s) \leq r - 1$, we have $f(\boldsymbol{x} + \chi_s) - f(\boldsymbol{x}) = \tilde{f}(T_{\boldsymbol{x}+\chi_s}) - \tilde{f}(T_{\boldsymbol{x}}) \geq \tilde{f}(T_{\boldsymbol{y}+\chi_s}) - \tilde{f}(T_{\boldsymbol{y}}) = f(\boldsymbol{y} + \chi_s) - f(\boldsymbol{y})$, which implies the diminishing return property of $\tilde{f}$. $\qquad\square$