[Reviews · NeurIPS 2015]

Submitted by Assigned_Reviewer_1

The paper proposes a modified version of the epsilon greedy algorithm for finding approximate solution to the generalized problem of submodular cover on the integer lattice. It is shown that the proposed algorithm provides a bicriteria approximation guarantee while having a running time which is polynomial in the input size.

The paper is among a class of resent papers that have been working on accelerating and hence scaling up the optimization methods to be practical for large modern data sets (as is commonly seen in machine learning tasks).

Summary: Overall, I think the paper is well-written and through and does a good job of providing theoretical guarantees for the proposed algorithms for submodular maximization over an integer lattice. The proposed problem has real world applications and the quality of the solution is shown through experiments on real and artificial datasets.

Submitted by Assigned_Reviewer_2

PAPER SUMMARY

The paper proposes a generalization of submodular cover to integer lattices. Specifically, the problem is to minimum a sub-additive function c over an integer lattice, under the constraint that the value of a DR-submodular function f is at least \alpha. The paper presents an efficient greedy algorithm, which provides a multiplicative bound on the objective c, and a multiplicative violation of the constraint f. Even in the case where c is DR-submodular, and the problem can be reduced from integer lattices to set functions, the worst-case runtime of the algorithm is better than the one suggested by this reduction.

COMMENTS

- The generalization of submodular cover is interesting, and nicely justified through some plausible applications.

- The paper is clearly written. However, section 1.1 contains several terms that are defined later. This makes it difficult to immediately follow the related work section that compares the guarantees of the proposed method with previous algorithms for special cases. Either all the terms should be specified in section 1.1, or the related work section should be moved to the end of the technical section.

- When c is DR-submodular, how does the theoretical guarantee obtained by the naive reduction compare to the proposed method? The paper only mentions that the proposed method is faster. Is it as accurate?

- Can multiple submodular constraints be handled using the proposed method?

- The functions c and f used in the experiments are hand-tuned, and their validity for the applications is not clear. However, for the setting used, the proposed method seems to perform more accurately than some baselines, and faster than others, thereby providing a desirable trade-off.

- Minor: equation (5): explain the notation "x*: optimal solution". By context, I think this means that "x* \in X*" where X* is the set of all optimal solutions. Is that right?
Summary: Interesting and useful generalization of submodular cover, with an efficient algorithm that provides provable guarantees. Some terms are used without definition. Comparison of theoretical guarantees with naive baseline is missing.

Submitted by Assigned_Reviewer_3

Overall I think the paper is well written.

I would like to see some connections of this notion of submodularity, to known ones on integer lattices, e.g. discrete concavity and convecity (Murota et al) and k-submodularity.
Summary: This paper studies the submodular set cover problem, and its generalization (submodular min with submodular cover constraints), under a certain generalization of submodular functions (or those which satisfy the diminishing returns property). The authors motivate this problem with several real world examples, and provide theoretical guarantees.

Author Feedback
Author rebuttal: First we would like to thank all the reviewers for their careful readings. We will address all the comments if the paper is accepted. Let us response to major comments in detail below.

To reviewer 2
-------------------------
- In our final version, we will improve our writing to avoid the usage of terms that are defined later.
- When c is DR-submodular, the theoretical approximation ratios of our method and the naive reduction coincide. Our method runs faster than the naive method with the same approximation guarantee.
- It is not straightforward to adapt our method to handle multiple submodular constraints. It is an interesting future work.

To reviewer 3
-------------------------
- From the view of discrete concavity, one can regard M-(natural)-concave functions as a tractable subclass of DR-submodular functions. k-submodularity is another generalization of submodularity (of a set function) for functions over D^S, where D = {d0, d1, ..., dk} is a finite domain. The definition of k-submodularity is symmetric over {d1, ..., dk}, and we do not have any natural ordering among {d1, ..., dk}. Hence, although it might be possible to identify D as the integer set {0, 1, ..., k}, we cannot deduce the diminishing return property used in our paper.

To reviewer 6
-------------------------
- We believe that the current experimental setting is the most basic and natural extension of existing models because unit cost corresponds to an independent chance of sensing.